# Autophagy and Apoptosis of Porcine Ovarian Granulosa Cells During Follicular Development

**DOI:** 10.3390/ani9121111

**Published:** 2019-12-10

**Authors:** Yuxin Zheng, Lizhu Ma, Ning Liu, Xiaorong Tang, Shun Guo, Bin Zhang, Zhongliang Jiang

**Affiliations:** 1College of Animal Science and Technology, Northwest Agriculture & Forestry University, Yangling 712100, China; yuxinzheng@nwsuaf.edu.cn (Y.Z.); malizhu@nwafu.edu.cn (L.M.); liuning19991225@sina.com (N.L.); xiaortang@126.com (X.T.); uniquewood@163.com (S.G.); 2College of Animal Science and Technology, State Key Laboratory for Sheep Genetic Improvement and Healthy Production, Xinjiang Academy of Agricultural and Reclamation Science, Shihezi 832000, China; 18309256612@163.com

**Keywords:** granulosa cells, follicular size, follicular atresia, autophagy, apoptosis

## Abstract

**Simple Summary:**

Granulosa cells (GCs) provide nutrients and information for oocytes in porcine follicles. Follicular atresia is closely related to both apoptosis and autophagy of granulosa cells in ovarian follicles; however, the follicular stages of granulosa cell apoptosis or autophagy during follicular development or atresia are poorly understood. We found that autophagy and apoptosis of GCs occurred in GCs from different size follicles during follicular development, and autophagy was mainly found in GCs of medium follicles, while apoptosis was mainly found in GCs of large follicles. These data provided some useful information to understand follicular atresia which is related to the fertility of sows.

**Abstract:**

Follicular atresia is closely related to both apoptosis and autophagy of granulosa cells (GCs) in ovarian follicles. In the present study, GCs were isolated from pig ovaries in small, medium and large antral follicles, and the current results showed that the proliferation of GCs was higher in medium follicles, and lower in large follicles compared to small follicles. The *Bax* and *Caspase 3* mRNA levels were significantly higher, but the ratio of *Bcl-2*/*Bax* was lower in GCs of large follicles. The marker genes of autophagy, *Atg3*, *Atg7* and *LC3* mRNA levels were higher in GCs from medium follicles. Apoptosis- and autophagy-related proteins had a similar expression pattern to the mRNA level. Our results showed that phosphorylated ERK (p-ERK) was activated in GCs of large follicles, while phosphorylated AKT (p-AKT) and phosphorylated mTOR (p-mTOR) were inhibited in GCs of medium follicles. Labeling of autophagic vesicles with 4’,6-diamidino-2-phenylindole (DAPI) and monodansylcadaverine (MDC) confirmed the results of gene transcription and protein expression in GCs of different size follicles. We conclude that autophagy and apoptosis of GCs occurred in different size follicles during follicular development, and autophagy was mainly found in GCs of medium follicles, while apoptosis was mainly found in GCs of large follicles.

## 1. Introduction

Many factors regulate the process of ovarian follicle development, which includes endocrine, paracrine and autocrine factors aimed to promote ovulation; however, only 1% or less of follicles in the ovary reach the preovulatory stage and about 99% of follicles undergo the degenerative process called atresia [1]. Although granulosa cell (GC) apoptosis is confirmed to be the main reason for follicular atresia [2], increasing evidence suggests that autophagy is induced in ovarian GCs during follicular atresia to promote apoptotic cell death by excessive self-digestion and degradation of essential cellular constituents [3,4]. Autophagy is a potential mechanism to alleviate cellular stress induced by hyperthermia in different follicular stages of the pig ovary [5]; nevertheless, the inhibition of transcription factor nuclear factor- κB (NF- κB) increased autophagy of porcine granulosa cells via JNK signaling and promoted steroidogenesis [6].

Autophagy and apoptosis are two inseparable processes. Autophagy is a process of degradation and recycling of cellular constituents resulting in constitutive, dynamic, evolutionarily conserved catabolism to maintain cell survival under various conditions of starvation, hypoxia, and interruption of growth signaling. However, the results of many studies suggest that autophagy promotes cell death and can be triggered by various stimuli that induce apoptosis [7,8]. In humans, the exposure of cells to oxidized low-density lipoprotein that induces endothelial cell apoptosis results in granulosa cell death by autophagy [9]. In rat GCs, serum starvation induced accumulation of autophagosomes to activate cell apoptosis through decreased Bcl-2 expression, which suggests that autophagy may promote GC apoptosis [10]. In mice, Follicle stimulating hormone (FSH)induces autophagy signaling in GCs via HIF-1α [11]. Therefore, autophagy may be involved in folliculogenesis, as granulosa cells are the primary site of apoptosis during follicle atresia [3].

Follicular atresia is not a redundant process and is in fact absolutely necessary for humans to maintain a healthy reproductive system [12]. During follicular growth and development, antral follicles are susceptible to atresia [13] and secondary or tertiary follicles are highly likely to become early atretic follicles if FSH is suppressed [14]. The GCs isolated from atretic follicles had higher susceptibility to cell apoptosis accompanied by higher expression of *FasL* mRNA in GCs, suggesting that follicular atresia may be regulated in a stage-specific manner [15]. A recent study demonstrated that the accumulation of autophagosomes induced apoptotic GC death through the decreased expression of Bcl-2 and the subsequent activation of caspases [3], suggesting that the autophagy of GCs leads to follicular atresia. In pig, although many of the recruited follicles in each estrous cycle proceed to ovulation, there remain many follicles destined for atresia. Therefore, analyses of GC autophagy are very helpful to understand the mechanism of follicle atresia in pigs. However, the stage-specific follicular atresia and signaling pathway that regulates GC autophagy during follicular development and/or atresia are not fully understood.

In the present study, we hypothesized that autophagy and apoptosis of porcine GCs occurred in different size follicles during follicular development. To test this hypothesis, the GCs were collected from small follicles (S-GCs), medium follicles (M-GCs) and large follicles (L-GCs), and marker genes and proteins of autophagy and apoptosis in GCs were investigated, respectively.

## 2. Materials and Methods

### 2.1. Ethics Statement

The present study was approved by the ethics committee of Northwest Agricultural and Forestry University, Shaanxi, China.

### 2.2. Granulosa Cell Isolation and Culture

The ovaries were obtained from adult pigs at a local abattoir, irrespective of the estrous cycle and transported to the laboratory in pre-warmed (37 °C) saline with 100 IU/mL penicillin and 100 μg/mL streptomycin. Follicles with clear and transparent fluid, slightly yellow follicular membranes and evenly distributed capillaries were healthy follicles. A 5 mL syringe with a 25-gauge needle was used to extract follicular fluid with granulosa cells from small follicles (follicle diameter less than 2 mm), medium follicles (follicle diameter between 2 and 6 mm) and large follicles (follicle diameter bigger than 6 mm). The cell suspension was then filtered through a 150 mesh steel sieve (Sigma-Aldrich, China) and centrifuged at 500× *g* for 10 min at room temperature. The cell sediment was diluted to 10 mL using fresh medium and the viability of GCs was assessed using the Trypan blue dye exclusion procedure.

To get the stable cells, GCs were cultured in serum-free medium that was conducive to the maintenance of both estradiol secretion and responsiveness to FSH. These responses were characterized using the procedures previously described [16], with a few modifications. Briefly, cells were cultured at a density of 1 × 10^6^/mL with DMEM/F12 containing sodium bicarbonate (10 mM), sodium selenite (4 ng/mL), bovine serum albumin (BSA) (0.1%, W/V, Sigma-Aldrich, St. Louis, MI, USA), penicillin (100 U/mL), streptomycin (100 μg/mL), transferrin (2.5 μg/mL), nonessential amino acid mix (1.1 mmol/L), insulin (10 ng/mL), androstenedione (10^−7^ M) and FSH (10 ng/mL, BIONICHE INC. Ottwa, ON, Canada). Cultures were maintained at 37 °C in 5% CO_2_ for 2 days. On day 2, GCs were harvested for further analysis.

### 2.3. Assessment of Cell Proliferation

A MTT assay kit was used to assess the proliferation of the GCs according to the manufacturer’s guidelines. Briefly, cells were cultured in 96-well plates (Corning Inc., Shanghai, China) at a density of 10^4^/100 μL. Four hours later, MTT solution (20 µL; 5 mg/mL) was added to each well and mixed gently. The cells with MTT solution were then incubated for 4 h at 37 °C and 5% CO_2_. A microplate reader (BioTek, Winooski, VT, USA) was used to read the absorbance of samples at 450 nm and 620 nm. The standard curve of absorbance related to viable cell number was prepared by culturing and measuring GCs at different plating densities in quintuplicate (from 1 × 10^3^ to 5 × 10^5^/200 μL) for 4 h. The number of viable cells/well was calculated based on a standard curve.

### 2.4. RNA Extraction and qRT-PCR

TRIzol reagent (Invitrogen Inc., Beijing, China) was used to extract total RNA from GCs according to the manufacturer’s instruction. Briefly, the culture medium was removed and the GCs were solubilized for 5 min at room temperature using 200 μL TRIzol in each well, and then transported into a 1.5 mL centrifuge tube. Then 0.2 mL of chloroform was added into the tube and mixed well, and the mixture was centrifuged at 12,000 g for 10 min at room temperature. The upper clear phase was transferred carefully to a fresh tube and 0.5 mL of isopropanol was added into the tube and mixed by vortex for 3 min. The mixture was centrifuged at 12,000 g for 10 min at 4 °C to precipitate the RNA. The pellet was resuspended immediately in 20 μL RNase-free water. The concentrations of RNA were quantified using a NanoDrop 2000 (Thermo, Waltham, MA, USA) at 260 nm absorbance. Reverse transcription PCR was performed on 1 μg of DNase-treated total RNA. The reaction mixture consisted of 1 mmol/L oligo-(dT) primer, 4 U Omniscript RTase, 0.25 mmol/L dNTP mix, and 19.33 U RNase inhibitor. The total reaction volume was 20 μL, and the samples were incubated at 37 °C for 1 h, and then terminated at 93 °C for 5 min.

Quantitative real-time PCR for gene expression was performed with SYBR Green I PCR Master Mix in a reaction volume of 20 μL using the ABI system (Applied Biosystems 7900, Warrington, UK). Common thermal cycling parameters (3 min at 95 °C, 40 cycles of 15 sec at 95 °C, 30 sec at 59 °C, and 30 sec at 72 °C) were used to amplify each transcript. Melting-curve analyses were performed to verify product identity. *Gapdh* was used as a housekeeping gene for normalization of the expression level of mRNA, the mRNA expression was calculated using ∆Ct = (Ct_mRNA_ − Ct_Gapdh_). All samples were assessed in triplicate. Porcine specific primers of target genes for qRT-PCR are listed in Table 1.

### 2.5. Western Blotting

Granulosa cells were harvested by using radioimmune precipitation assay lysis buffer (RIPA buffer, Pierce Chemical, IL, USA) and protein was quantified by the BCA method (Pierce, Chemical, IL, USA). Cell lysates containing 15 μg total protein were separated by sodium dodecyl sulfate polyacrylamide gel electrophoresis (SDS-PAGE, 10% acrylamide gel), and the proteins were transferred to polyvinylidene difluoride membranes (PVDF, Millipore, Billerica, MA, USA). After blocking with 5% BSA in Tris buffered saline (10 mM Tris, 150 mM NaCl, pH 7.4) containing 0.1% Tween-20 (TBS-T) for 2 h, membranes were incubated with primary antibodies in TBST overnight at 4 °C. The antibodies of BAX (1: 2000, # SC-20067), BCL2 (1: 2000, # SC-23960), Caspase 3 (1: 1000, # SC-56053), ATG3 (1: 1000, # SC-393660), ATG7 (1:500, # SC-376212), LC3 (1:2000, # SC-398822), mTOR (1:1000, # SC-517464), p-mTOR (ser2448, 1:1000, # SC-293132) and β-actin (1:5000, # KM9001T) were obtained from Santa Cruz Biotechnology (Santa Cruz, USA); AKT (1:3000, # 9272). The antibodies of p-AKT (1:3000, # 4060), ERK1/2 (1:2000, # 4695) and p-ERK1/2 (1:2000, # 4370) were obtained from Cell Signaling Technology (Cell Signaling, Danvers, MA, USA). Subsequently, the membranes were washed three times in TBST and incubated for 2 h at room temperature with anti-rabbit HRP-conjugated IgG (1:5000, # LK2001, Sungene Biotechnology, Tianjin, China) diluted in TBST with 5% BSA. The protein bands were labeled on the membranes using an enhanced chemiluminescence (ECL, Millipore, Danvers, MA, USA) detection system. Semiquantitative analysis was performed using the Image J software (NIH, Bethesda, MD, USA).

### 2.6. Assessment of Cell Apoptosis

The percentage of apoptotic GCs were analyzed by flow cytometry with an Annexin V-FITC apoptosis detection kit according to the manufacturer’s instructions. Briefly, after 2 days culture, GCs were transferred into a sample tube and were washed twice in PBS, then resuspended in 400 μL binding buffer. Then, 5 μL Annexin V-FITC and 5 μL propidium iodide (PI) were added into the cell suspensions and were incubated for 15 min at room temperature, protected from light. The single-stained cells of Annexin V-FITC or PI were analyzed to adjust fluorescence compensation to remove spectral overlap. The rate of apoptotic GCs was analyzed using a FACS (Becton Dickinson, San Jose, CA, USA). The maximum excitation wavelength of FITC was 488 nm, and the maximum emission wavelength was 525 nm. FlowJo software (Treestar, Inc., San Carlos, CA, USA) was used for analysis.

### 2.7. Labeling of Autophagy and Apoptosis

The formation of autophagic vesicles is a marker of cell autophagy, which is labeled by monodansylcadaverine (MDC) and appeared as bright green punctate under fluorescence microscopy, and 4’,6-diamidino-2-phenylindole (DAPI) was used to label DNA in the nucleus, which appeared as a blue fluorescence. Briefly, after 2 days of culture, the cells were incubated with 0.02 mM MDC and/or DAPI (1 μg/mL) in 24-well plates at 37 °C for 20 min, immediately followed by fixation in 4% paraformaldehyde for 30 min at 4 °C. A fluorescence microscope (excitation: 390 nm, emission: 460 nm) was used to analyze the cells with bright punctate. Two hundred cells per sample were analyzed and the percentage of autophagic cells was calculated.

### 2.8. Statistical Analysis

Data were analyzed by one-way analysis of variance (ANOVA) and Duncan’s test using GraphPad Prism 6 software (GraphPad Software, V6, La Jolla, CA, USA). Data are presented as the means ± SEM of at least three independent replicates. Differences with *p* values < 0.05 were considered statistically significant.

## 3. Results

### 3.1. Proliferation and Apoptosis of GCs in Follicles

Proliferation of GCs was detected by MTT assay kit (Cayman chemical company, Ann Arbor, MI, USA) (Figure 1A). Compared to small follicles, GC proliferation was higher in medium follicles, while lower in large follicles (*p* < 0.05). The apoptotic GCs in follicles were detected using flow cytometry with an Annexin V-FITC apoptosis detection kit. The results indicated that the percentage of apoptotic cells in small and medium follicles was 25.4% and 25.5%, respectively, which were lower than that of large follicles (33.54%, *p* < 0.05, Figure 1B). The percentage of apoptotic cells did not differ between cells from small and medium follicles. These results indicated that the proliferation and apoptosis of GCs occurs in different stages of follicular growth.

### 3.2. Relative mRNA Level in GCs of Different Size Follicles

To study the situation of GCs in different size follicles, relative mRNA level of apoptosis-related and autophagy-related genes was quantified by qRT-PCR (Figure 2 and Figure 3). The transcriptional activity of *Bcl-2,* an anti-apoptotic gene in GCs isolated from large follicles was significantly lower than those of small and medium follicles (*p* < 0.01, Figure 2A); however, *Bax,* a pro-apoptotic gene showed the strongest transcriptional activity in GCs of large follicles (*p* < 0.01, Figure 2B). The ratio of *Bcl2* to *Bax* is an important indicator to determine whether the cells are apoptotic or not. As shown in Figure 2C, the ratio of *Bcl2*/*Bax in* GCs of large follicles was lower than those of small and medium follicles, indicating that the GCs in large follicles had a higher ratio of apoptotic cells. In the present study, the mRNA level of *Caspase 3*, an executor of apoptosis was significantly higher in GCs of large follicles than those of small and medium follicles (Figure 2D).

As key factors of autophagic vesicle formation, ATG3 and ATG7 can promote the transformation of LC3 I into LC3 II, resulting in activation of autophagy. In the present study, the relative mRNA level of *ATG3* in medium follicular GCs was significantly higher than that in small and large follicular GCs (*p* < 0.01, Figure 3A). Compared with small follicular GCs, the relative expression of *ATG7* increased significantly in medium and large follicular granulosa cells (*p* < 0.05, Figure 3B). The mRNA level of *LC3* in GCs of medium follicles was significantly higher than those of small and large follicles (*p* < 0.01, Figure 3C) and there was no difference in *LC3* mRNA in GCs between small and large follicles. This result showed that the transcription level of apoptosis-related genes was higher in GCs of large follicles, while the autophagy-related genes were higher in GCs of medium follicles.

### 3.3. Protein Expression of Autophagy and Apoptosis in GCs

To confirm the results of gene transcription, the expression of autophagy-related and apoptosis-related proteins in GCs was detected (Figure 4A and Figure 5A). It can be seen that BCL2, an anti-apoptotic protein is expressed at a significantly higher level in GCs of small and medium follicles than that of large follicles (*p* < 0.05, Figure 4B). On the contrary, the expression of BAX, an apoptotic protein was significantly lower in small and medium follicular GCs relative to large follicles (*p* < 0.05, Figure 4C). Caspase 3, an apoptosis executor, was expressed at a higher level in large follicular GCs than those of small and medium follicular GCs (*p* < 0.05, Figure 4D).

A similar protein expression of ATG3, ATG7 and LC3 was observed, with all proteins having higher levels in medium follicles, compared with small and large follicles (*p* < 0.05, Figure 5B–D). Meanwhile, expression of these proteins did not differ between GCs from small and large follicles.

### 3.4. Labeling of Autophagosomes and the Nucleus in GCs

In this experiment, the autophagosomes and nucleus were labeled by MDC and DAPI, respectively. As shown in Figure 6, the blue fluorescence indicates the nucleus and the green fluorescence indicates the autophagosomes. The current results showed that GCs from medium size follicles contained a greater number of autophagosomes. In GCs of large follicles, DNA fragments were labeled by DAPI staining. This result of fluorescence labeling also confirmed autophagy in GCs of medium follicles and apoptosis in GCs of large follicles.

### 3.5. Activation of Autophagy and Apoptosis Signaling in GCs

In order to understand the regulation of autophagy and apoptosis of GCs better during follicular development, the phosphorylated mTOR (p-mTOR), AKT (p-AKT) and ERK1/2 (p-ERK1/2) were determined in this experiment. Our results showed similar expression patterns of mTOR, AKT and its phosphorylation in GCs of different size follicles. The p-mTOR was decreased significantly in GCs of medium and large follicles (*p* < 0.05, Figure 7A), and there was no difference in p-mTOR in GCs between medium and large follicles. A significantly lower p-AKT was observed in GCs of medium follicles than those of small and large follicles (*p* < 0.05, Figure 7B). However, a significantly higher level of p-ERK was found in large follicles than those of small and medium follicles (Figure 7C). These results showed that the phosphorylation of mTOR and AKT might be involved in the activation of autophagy in GCs of medium follicles.

## 4. Discussion

One of the main functions in ovaries is to generate oocytes from thousands of follicles, however, about 1% follicles are “lucky” enough to grow to ovulate and 99% of follicles breakdown by a process known as follicular atresia [17]. Although it is well documented that apoptosis of granulosa cells is the main reason for follicular atresia [18], an increasing number of pathways and cellular processes such as autophagy have also been found to be involved in this regulation of follicular atresia [12]. Here, we demonstrated that autophagy and apoptosis occur in GCs in different stages of follicles and autophagy mainly occurs in GCs of medium follicles, while apoptosis mainly occurs in GCs of large follicles. However, autophagy was found in bovine granulosa cells of large follicles, while apoptosis was found in medium follicles [19]. These results showed variation between species.

A study of follicle culture in vitro for 24 h showed that the apoptotic rate of GCs from small follicles was higher than those of large and medium follicles without FSH supplementation; however, the percentage of apoptotic GCs was significantly inhibited by addition of 3 or 5 IU/mL FSH [15]. This result indicated that FSH plays an important role in GCs to prevent GCs from apoptotic injury during follicle development. In the present study, the results showed that the proliferation of GCs in medium follicles was significantly higher than that of cells in small and large follicles. In addition, GC proliferation in small follicles was significantly higher than that of large follicles. Moreover, when GCs were cultured in vitro for 48 h, the percentage of GC apoptosis in large follicles was the highest in the three follicle sizes with 10 ng/mL FSH addition. A possible explanation for these results is that the GCs in medium follicles were capable of proliferating and keeping follicles growing well in the presence of FSH. Nevertheless, the decreased GC proliferation in large follicles showed that GCs might degenerate or undergo apoptosis, resulting in follicular atresia. This speculation was confirmed by the transcription level of apoptosis-related genes and protein expression. The current results suggest that GCs in large follicles undergo apoptosis and follicular atresia even with the addition of FSH. Multiple bits of evidence suggest that autophagy has a number of different functions including protection from stress, starvation, immune regulation, differentiation, proliferation, cell death, and the regulation of reproduction [7,8,9]. In the ovary, follicular development is significantly altered by autophagy in follicles and systemic metabolic conditions with profound consequences on reproductive outcomes [12]. It was demonstrated that autophagy was downregulated in rat GCs in the presence of FSH [10] and FSH promotes the activation of autophagy via upregulation of HIF-1α in MGCs [11]. A key autophagy-related protein, LC3, was present in primary GCs and pre-antral follicles but absent in primordial follicle cells and oocytes [3]. In humans, when exposed to oxidized low-density lipoprotein (OxLDL), GC death was induced by cell autophagy, leading to follicular atresia. In the present study, *Atg3*, *Atg7* and *LC3* mRNA levels were measured because the cytosolic form of LC3 (LC3-I) is recruited to autophagosomal membranes [20], and the activity of ATG3 is involved in the formation of LC3-II on the surface of autophagosomes by binding to ATG7 [12]. The current results indicated that autophagy was mainly occurring in GCs isolated from medium follicles.

Autophagy and apoptosis were regulated by various microenvironments and stimuli. Extracellular signal-regulated kinase (ERK) controls many aspects of cell physiology including cell apoptosis [21]. AKT is a known regulator of autophagy [22] and the AKT/mTOR pathway is one of the major signaling pathways involved in autophagy [23]. Here, we further showed signaling pathways of ERK, AKT, mTOR and their phosphorylation in GCs of different size follicles. In the present study, the phosphorylated ERK (p-ERK1/2) in GCs from large follicles was significantly higher than that of GCs from small and medium follicles, which demonstrates that ERK1/2 phosphorylation was increased significantly in apoptotic cells in previously published studies [24,25]. Furthermore, MAPK/ERK1/2 and PI3K/AKT are two signaling pathways that negatively regulate autophagy through the mTOR pathway [26,27]. The mTOR signaling pathway inhibits autophagy when mTOR kinase expression increases with increasing cellular amino acid, ATP, and hormone concentrations, which act as a negative regulator of autophagy [28,29]. The PI3K/AKT signaling pathway also resulted in activation of the mTOR signaling pathway in cancerous and neuronal cells to inhibit autophagy [30,31]. Here, lower levels of p-AKT and p-mTOR in medium follicles indicated that the autophagy of GCs was induced through the inhibition of the phosphorylation of the AKT/mTOR signaling pathways.

In the process of autophagy, the autophagosomes presented as a granular distribution throughout the cell, however, in the late stages of apoptosis, the nuclear membrane was destroyed and DNA fragments were found in the nucleus. In this experiment, double staining of MDC and DAPI was used to confirm the autophagy and apoptosis in GCs during follicular development and the current results indicated that autophagy of GCs in medium follicles has a higher level than GCs in small and large follicles. Meanwhile, the nucleus of large follicular granulosa cells was notably disrupted.

## 5. Conclusions

The present study demonstrated that autophagy and apoptosis of GCs occurred in different size follicles during follicular development, and autophagy was mainly found in GCs of medium follicles, while apoptosis was mainly found in GCs of large follicles. These data provide some useful information to understand follicular atresia, which is related to sow fertility. The factors regulating induction and the mechanisms involved in autophagy and/or apoptosis in porcine ovarian follicles require further studies.

## Figures and Tables

**Figure 1 animals-09-01111-f001:**
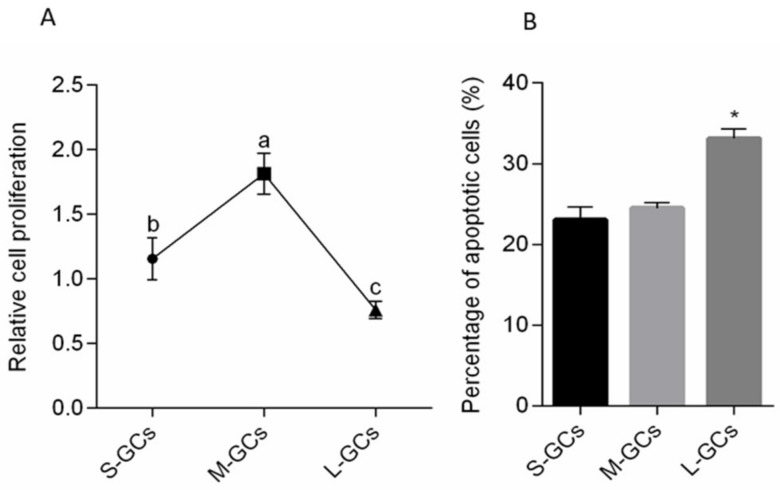
Proliferation and apoptosis of granulosa cells following 2 days of culture. The MTT assay kit was used to assess cell proliferation (**A**), and flow cytometry with an Annexin V-FITC apoptosis detection kit was used to assess the apoptotic GCs (**B**). Data are means ± SEM from three independent replicates. Different letters (**A**) and asterisks (**B**) in the diagram indicate a significant difference (* *p* < 0.05). S-GCs, GCs isolated from small follicles; M-GCs, GCs isolated from medium follicles; L-GCs, GCs isolated from large follicles.

**Figure 2 animals-09-01111-f002:**
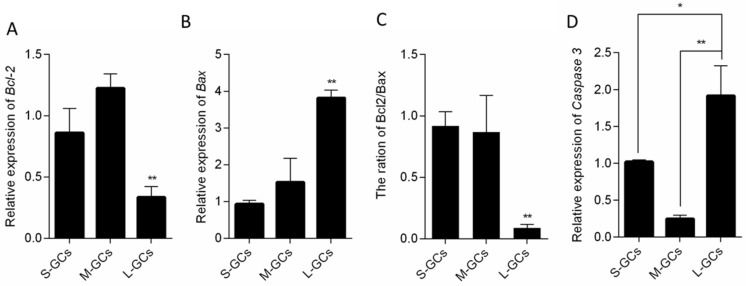
Relative mRNA level of apoptosis-related genes in GCs. Relative mRNA level of *Bcl2* (**A**), *Bax* (**B**), the ratio of *Bcl2/Bax* (**C**) and *Caspase 3* (**D**). Data are means ± SEM from three independent replicates. * *p* < 0.05, ** *p* < 0.01. S-GCs, GCs isolated from small follicles; M-GCs, GCs isolated from medium follicles; L-GCs, GCs isolated from large follicles.

**Figure 3 animals-09-01111-f003:**
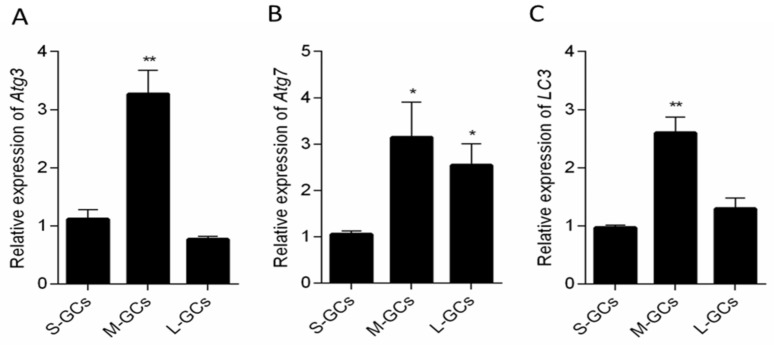
Relative mRNA level of autophagy-related genes in GCs. Relative mRNA level of *Atg3* (**A**), *Atg7* (**B**), and *LC3* (**C**). Data are means ± SEM of three independent replicates. * *p* < 0.05, ** *p* < 0.01. S-GCs, GCs isolated from small follicles; M-GCs, GCs isolated from medium follicles; L-GCs, GCs isolated from large follicles.

**Figure 4 animals-09-01111-f004:**
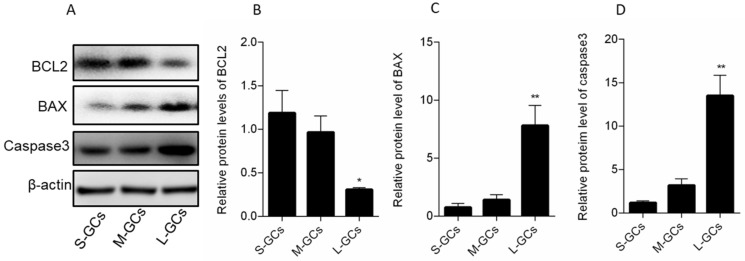
Apoptosis-related protein expression in GCs of follicles (**A**). Relative protein levels of BCL2 (**B**), BAX (**C**) and Caspase 3 (**D**). Data are means ± SEM of three independent replicates. * *p* < 0.05, ** *p* < 0.01. S-GCs, GCs isolated from small follicles; M-GCs, GCs isolated from medium follicles; L-GCs, GCs isolated from large follicles.

**Figure 5 animals-09-01111-f005:**
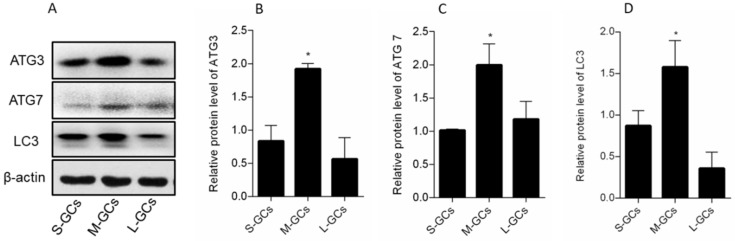
Autophagy-related protein expression in GCs of follicles (**A**). Relative protein levels of ATG3 (**B**), ATG7 (**C**) and LC3 (**D**). Data are means ± SEM of three independent replicates. * *p* < 0.05. S-GCs, GCs isolated from small follicles; M-GCs, GCs isolated from medium follicles; L-GCs, GCs isolated from large follicles.

**Figure 6 animals-09-01111-f006:**
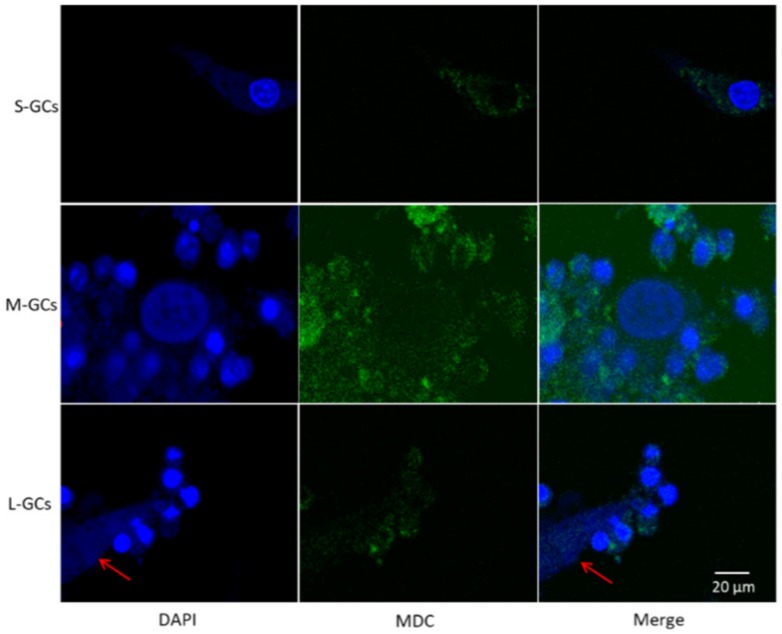
Double staining of DAPI and MDC to label the autophagosomes and GC nucleus. The red arrow indicated the DNA fragments in GCs of large follicles. Scale bars correspond to 20 μm. S-GCs, GCs isolated from small follicles; M-GCs, GCs isolated from medium follicles; L-GCs, GCs isolated from large follicles.

**Figure 7 animals-09-01111-f007:**
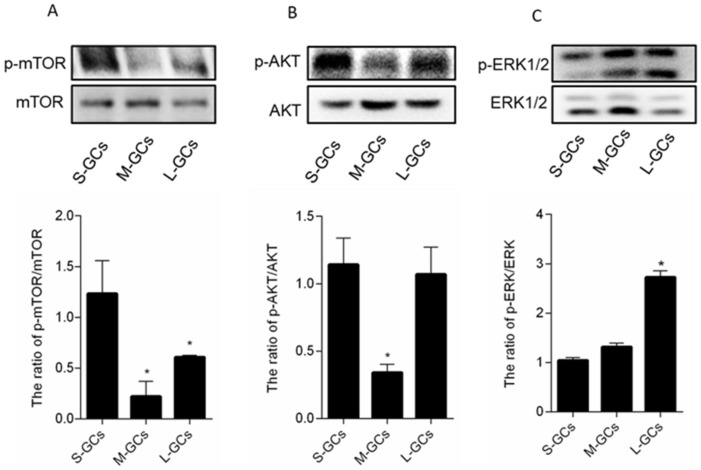
Protein expression of the autophagy and apoptosis signaling pathways activated in GCs during follicular development. (**A**) mTOR and p-mTOR; (**B**) AKT and p-AKT; (**C**) ERK1/2 and p-ERK1/2. Data are means ± SEM of three independent replicates. * *p* < 0.05. S-GCs, GCs isolated from small follicles; M-GCs, GCs isolated from medium follicles; L-GCs, GCs isolated from large follicles.

**Table 1 animals-09-01111-t001:** Swine specific primer sequences for qRT-PCR.

Gene	(5′–3′) Forward Primer	(5′–3′) Reverse Primer	Amplicon Size (bp)
*Gapdh*	GGACTCATGACCACGGTCCAT	TCAGATCCACAACCGACACGT	220
*Bcl2*	AGAGCCGTTTCGTCCCTTTC	GCACGTTTCCTAGCGAGCAT	231
*Bax*	ATGATCGCAGCCGTGGACACG	ACGAAGATGGTCACCGTCTGC	296
*Caspase3*	TTGGACTGTGGGATTGAGACG	CGCTGCACAAAGTGACTGGA	165
*Atg3*	CACGACTATGGTTGTTTGGCTATG	GGTGGAAGGTGAGGGTGATTT	127
*Atg7*	AGATTGCCTGGTGGGTGGT	GGGTGATGCTGGAGGAGTTG	1021
*LC3*	AGGTATTCAGTCACCTTTGTTTCA	GAAACGGGCTCTGCAGTCTA	192

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
