# Peer review of "Autophagy and Apoptosis of Porcine Ovarian Granulosa Cells During Follicular Development"

_animals, 2019, doi:10.3390/ani9121111_

Round 1
Reviewer 1 Report
This study has examined markers of apoptosis and autophagy in granulosa cells collected from small, medium and large ovarian follicles in the pig. The results provide evidence for autophagy in granulosa cells from medium follicles, while apoptosis was mainly found in large follicles. The cells were collected from the follicles and cultured for 2 days prior to analysis. This makes it unclear as to how accurately the gene and protein expression reflect in vivo levels, or whether cells from different size follicles may respond differently to the culture environment. There is some discussion of of the influence of FSH in the culture medium, but whether optimal culture conditions could vary between cells from different follicle stages is not clear.
The authors have recently published similar experiments in bovine granulosa cells that identify the highest levels of autophagy markers in cells from large follicles, but these results are not discussed. These results appear relevant, given that they differ from the current study, suggesting variation between species. Overall there is a lack of commentary on the relevance of studying the pig ovary, why the study was performed in porcine granulosa cells, and why the results observed in porcine cells differ from bovine cells. Lower levels of FSH were also used for the studies with bovine cells, was there a reason for this?
Specific comments:
The title should indicate that the cells are porcine.
Abstract: Throughout the manuscript the authors need to be careful not to describe the results in such a way that implies they have looked at the markers within follicles. The results have been generated from cells from small, medium or large follicles. For example, the sentence "The marker genes of autophagy, Atg3, Atg7 and LC3 mRNA levels were higher in medium follicles" implies that analysis has been performed on the follicles. In fact, the study utilises cells from medium follicles and examines them following 2 days of culture. This sentence should refer to "mRNA levels were higher in cells from medium follicles."
Line 28: is it correct that AKT and mTOR were activated in granulosa cells from medium follicles? Their levels were lower in these cells.
Line 29-30: the sentence "The fluorescence labeling of DAPI and MDC confirmed the results of gene transcription and protein expression in GCs of different size follicles" provides the reader with little information, as the purpose of these stains is not evident in the Abstract.
Introduction: The introduction has limited information on why the study was performed in porcine granulosa cells. The pig ovary and porcine granulosa cells are mentioned in one sentence (lines 43-46), but there is no other explanation of why pig ovaries were chosen, and there is no mention of species in the hypothesis and aim paragraph. The introduction should include an explanation for why porcine ovaries were the focus of the study.
Methods: It is unclear whether apoptosis has been measured following 48 hours of culture, or immediately following removal of the cells from the follicle. The methods description suggests that cells were assessed following collection, while the Fig 1 legend indicates apoptosis was assessed after 2 days culture. Similarly, it is not fully clear in the methods when the MDC and DAPI staining was performed.
Is three independent replicates based on three separate experiments (3 different collections of ovaries, on different days), or is this 3 culture dishes?
Results: As indicated in the Abstract care needs to be taken to indicate that cells from small, medium and large follicles have been used. For example, the sentence "There was no difference of apoptotic cell percentage in both small and medium follicles: (lines 179-180), would more accurately be stated as "The percentage of apoptotic cells did not differ between cells from small and medium follicles".
For Figures 2, 3, 5, 6, 7 the description of the P symbol needs to be clearer. Is this that bars with the symbol differ significantly from bars without the symbol. The in-text description of these differences is clear, but the symbol use within the Figure and the Fig legend are not clear.
The Figure number is missing from Figs 4-7.
The mRNA and protein expression for Caspase 3 differ, with lowest protein abundance in cells from small follicles, but lowest mRNA in cells from medium follicles. Is there an explanation for this?
Line 261: The sentence "These results show that the autophagy was activated in GCs of medium follicles" is not clear. These results describe signalling pathways that may be involved in the activation of autophagy, rather than providing evidence of autophagy?
Minor comments:
Line 16: found in
Line 17: found in
Lines 18-19: this sentence is difficult to follow and requires rewording.
Line 26: similar expression pattern
Line 32: found in
Line 33 found in
Line 38: or less of follicles (please insert the word of)
Line 39: 99% of follicles
Line 41: increasing evidence suggests
Line 44: follicular stages of the pig ovary
Line 52-53: granulosa cell death
Line 62: accompanied by higher expression
Line 79: in pre-warmed
Line 82: from small follicles
Line 122-123: please check this information, there are letters and numbers that are not clear
Line 139: please delete the words "were from" as they have been repeated.
Line 194: however, Bax
Line 219: autophagy-related
Line 233-234: Meanwhile expression of these proteins did not differ between GCs from small and large follicles.
Line 260: found
Line 269: 99% of follicles
Line 273: are occurring in
Line 274: and autophagy mainly occurs in GCs of medium follicles, while apoptosis mainly occurs in GCs of large follicles.
Line 320: found in the nucleus
Line 327: found in
Line 328: found in
Line 329: which is related to sow's fertility. The factors regulating induction and the mechanisms involved in autophagy and/or apoptosis in porcine ovarian follicles require further study.
Reviewer 2 Report
The authors examined the ratio of autophagy and apoptosis in granulosa cells (GCs) derived from different stages of follicles. In their results, the autophagy was activated in GCs of medium follicles. The p-mTOR was decreased significantly in GCs of medium and large follicle, and there was no difference of p-mTOR in GCs between medium and large follicles. A significantly lower p-AKT was observed in GCs of medium follicles than those of small and large follicles. In the results of apoptosis, the expression of BAX was significantly lower in small and medium follicular GCs relative to large follicles and Caspase 3 was expressed at a higher level in large follicular GCs than those of small and medium follicular GCs. The experiment design and review were well written. In addition, the results were presented in accordance with their hypothesis, and this article is enough to be accepted for publication.
Author Response
thanks
Reviewer 3 Report
The manuscript "Autophagy and apoptosis of ovarian granulosa cells during follicular development” brings a very interesting information on atresia in the porcine ovarian follicles during follicular development. Follicular atresia is closely related to both apoptosis and autophagy of granulosa cells. However, the expression pattern of genes related to both processes, depends on the size of antral follicles.
The work is well-designed and performed but the manuscript requires the correction including language. Some examples which need to be corrected
line 32 : “was founded”
line. 138: “... were obtained from were from Santa Cruz Biotechnology...”
line 182: why authors use different letters in fig. 1A and asterisk in fig. 1B.
line 182: description in the figure “folds of cell proliferation” should be changed. How was the fold of cells calculated?
line188: “indicated significantly different” to be corrected
line 196 “..the cells are apoptosis...”
line 3299 “..is related sow’s fertility..”
Authors conclude that results presented in the current manuscript put more light on the factors which are involved in processes of follicular atresia. In my opinion, the authors should have tried to find interaction between these two ways which lead to cell elimination.
This study shows that both autophagy and apoptosis are involved in granulosa cells death, however the level of activity of these processes differs depending on the size of the follicle. Coimmunoprecipitation experiment should allow to find whether crosstalk between autophagy and apoptosis exists.
Other paper which regrettably was not cited (Meng et al. Biol Reprod, 2019) demonstrated the different cell death pathways via autophagy in preantral rat follicles and via apoptosis in antral ones.
Author Response
Author Response for reviewer 3
Comments and Suggestions for Authors
The manuscript "Autophagy and apoptosis of ovarian granulosa cells during follicular development” brings a very interesting information on atresia in the porcine ovarian follicles during follicular development. Follicular atresia is closely related to both apoptosis and autophagy of granulosa cells. However, the expression pattern of genes related to both processes, depends on the size of antral follicles.
The work is well-designed and performed but the manuscript requires the correction including language. Some examples which need to be corrected
line 32 : “was founded”
Answer: it was revised to “was found in” thanks.
line. 138: “... were obtained from were from Santa Cruz Biotechnology...”
Answer: “were from” was deleted, thanks.
line 182: why authors use different letters in fig. 1A and asterisk in fig. 1B.
Answer: In figure 1A, the letters were used to show the different among the granulosa cells from small, media and large follicles, the different are existed in the cells of three type follicles. In figure 1 B, the different are only existed in the cells from large follicle and media, small follicles, so we use the asterisk to show the different.
line 182: description in the figure “folds of cell proliferation” should be changed. How was the fold of cells calculated?
Answer: it was revised to “Relative cell proliferation”. In this experiment, the cells proliferation was assessed by MTT assay kit
line188: “indicated significantly different” to be corrected
Answer: It was revised to “indicated significantly difference”, thanks.
line 196 “..the cells are apoptosis...”
Answer: It was revised to “the cells apoptosis”, thanks
line 3299 “..is related sow’s fertility..”
Answer: It was revised to “is related to sow’s fertility”.
Authors conclude that results presented in the current manuscript put more light on the factors which are involved in processes of follicular atresia. In my opinion, the authors should have tried to find interaction between these two ways which lead to cell elimination.
Answer: Thanks, This is a very good suggestion, we are planning to perform this experiment next step.
This study shows that both autophagy and apoptosis are involved in granulosa cells death, however the level of activity of these processes differs depending on the size of the follicle. Coimmunoprecipitation experiment should allow to find whether crosstalk between autophagy and apoptosis exists.
Answer: This experiment has been finish in our lab, we put this result into another manuscript. Thanks.
Other paper which regrettably was not cited (Meng et al. Biol Reprod, 2019) demonstrated the different cell death pathways via autophagy in preantral rat follicles and via apoptosis in antral ones.
Answer: Thanks, we will be more careful and take our eyes on more papers next time.
Round 2
Reviewer 1 Report
The majority of the comments have been addressed.
Some further comments:
Line 17-18. The final sentence is unclear, the words "related GCs autophagy and/or apoptosis in ovarian follicles." could be deleted.
Line 65: The added sentence "In pig, a number of follicles are atresia even multiple ovulations in the ovary" is unclear and requires rewording.
Line 149-152: This section still reads as though GCs were isolated from the follicles, filtered, diluted, transferred to a sample tubes and then apoptosis was measured. It is not clear in this section that cells were cultured for 48 hours between collection and analysis.
Line 176: "was detected using" please correct.
Line 195: "whether the cells are apoptotic or not." please correct.
Line 196 states that Bcl2/Bax ratio is higher in large follicles. Please check this statement.
Figures 2 and 3 do not have A, B, C, D labels.
Line 230-231 should read " A similar protein expression of ATG3, ATG7 and LC3 was observed, with all proteins having higher levels in medium follicles, compared with small and large follicles (P<0.05, Fig. 5B, 5C, 5D).
Line 303: "was mainly occurring in" please correct.
Author Response
The majority of the comments have been addressed.
Some further comments:
Line 17-18. The final sentence is unclear, the words "related GCs autophagy and/or apoptosis in ovarian follicles." could be deleted.
Answer:It was deleted.
Line 65: The added sentence "In pig, a number of follicles are atresia even multiple ovulations in the ovary" is unclear and requires rewording.
Answer: it was revised to “In each estrus cycle of swine, tens of follicles ovulate in the ovary, however, there still have plenty of follicles atresia.”
Line 149-152: This section still reads as though GCs were isolated from the follicles, filtered, diluted, transferred to a sample tubes and then apoptosis was measured. It is not clear in this section that cells were cultured for 48 hours between collection and analysis.
Answer: It was revised to “Briefly, after 2 days culture, GCs were transferred into a sample tube and were washed twice in PBS ,……..”.
Line 176: "was detected using" please correct.
Answer: It was revised to “was detected by”
Line 195: "whether the cells are apoptotic or not." please correct.
Answer: It was revised.
Line 196 states that Bcl2/Bax ratio is higher in large follicles. Please check this statement.
Answer: It was revised to “Bcl2/Bax ratio is lower.”
Figures 2 and 3 do not have A, B, C, D labels.
Answer: It was labeled and revised.
Line 230-231 should read " A similar protein expression of ATG3, ATG7 and LC3 was observed, with all proteins having higher levels in medium follicles, compared with small and large follicles (P<0.05, Fig. 5B, 5C, 5D).
Answer: It was revised.
Line 303: "was mainly occurring in" please correct.
Answer: it was revised